# Ion Drift and Polarization in Thin SiO_2_ and HfO_2_ Layers Inserted in Silicon on Sapphire

**DOI:** 10.3390/nano12193394

**Published:** 2022-09-28

**Authors:** Vladimir P. Popov, Valentin A. Antonov, Andrey V. Miakonkikh, Konstantin V. Rudenko

**Affiliations:** 1Rzhanov Institute of Semiconductor Physics SB RAS, 13 Lavrentiev Avenu, 630090 Novosibirsk, Russia; 2Valiev Institute of Physics and Technology RAS, 36 Nakhimovsky Avenu, bld.1, 117218 Moscow, Russia

**Keywords:** silicon-on-sapphire, hafnia, alumina, interlayers

## Abstract

To reduce the built-in positive charge value at the silicon-on-sapphire (SOS) phase border obtained by bonding and a hydrogen transfer, thermal silicon oxide (SiO_2_) layers with a thickness of 50–310 nm and HfO_2_ layers with a thickness of 20 nm were inserted between silicon and sapphire by plasma-enhanced atomic layer deposition (PEALD). After high-temperature annealing at 1100 °C, these layers led to a hysteresis in the drain current–gate voltage curves and a field-induced switching of threshold voltage in the SOS pseudo-MOSFET. For the inserted SiO_2_ with a thickness of 310 nm, the transfer transistor characteristics measured in the temperature ranging from 25 to 300 °C demonstrated a triple increase in the hysteresis window with the increasing temperature. It was associated with the ion drift and the formation of electric dipoles at the silicon dioxide boundaries. A much slower increase in the window with temperature for the inserted HfO_2_ layer was explained by the dominant ferroelectric polarization switching in the inserted HfO_2_ layer. Thus, the experiments allowed for a separation of the effects of mobile ions and ferroelectric polarization on the observed transfer characteristics of hysteresis in structures of Si/HfO_2_/sapphire and Si/SiO_2_/sapphire.

## 1. Introduction

Silicon-on-sapphire (SOS) substrates with silicon device nanolayers are a promising material for the next generation (5G) of mobile phone “digital radio” chips, but the structural perfection of the thin silicon layers in them obtained by different methods is still worse than that of ultrathin layers of silicon-on-insulator (SOI) structures, such as HR-TR SOI [1,2,3]. There have been few reports of the SOS structure being produced using the hydrogen-induced thin Si layer transfer on sapphire by wafer bonding [3,4,5,6,7]. Usually, sapphire bonding is used for nitride semiconductors or niobate dielectric wafers with coefficients of thermal expansion (CTE) similar to that of sapphire. To overcome the CTE difference, the authors [3] used local fast laser heating to diminish the full stresses in the bonded wafers and obtained only partial Si layer transfer. Instead, in [4,5,6,7], we suggested silicon and sapphire wafer bonding at a moderate temperature, followed by hydrogen-induced thin Si layer transfer on the whole sapphire wafer at an elevated temperature. We developed a method similar to SmartCut© by implanting a hydrogen-induced Si layer transfer in vacuum at an elevated temperature [4,5,6,7]. The need for high-temperature annealing of implantation defects has led to the growth of a nanometer-thin silicon oxide interlayer, formed as a result of silicon oxidation by decomposed water molecules at the bonding interface and alumina reduction by silicon atoms. Under these conditions, a large built-in positive charge is formed at the SOS structure interface [5]. During measurements, this charge caused a large negative threshold voltage, V_g_ = V_t,e_, for the current I_ds_ in the pseudo-MOSFET channel. With a sapphire substrate thickness of 100 µm, the threshold gate voltage V_t,e_ exceeds −10 kV. To reduce the effect of this charge on the pseudo-MOSFET characteristics, we proposed inserting SiO_2_ layers with a thickness of 300 nm embedded between the device silicon layer and sapphire, and that provides V_t,e_ > −0.5 kV [3,4,5]. Even SiO_2_ layers with a thickness of 50 nm gave V_t,e_ < −6.0 kV, which was outside the available gate bias voltage range V_g_ = ±6 kV. However, a thick SiO_2_ layer dramatically reduces the heat sink from the silicon device layer [8]. Moreover, there is a problem with choosing another inserted dielectric layer between the semiconductor and sapphire due to the large lattice and CTE mismatches [9]. Use of the PEALD Al_2_O_3_ interlayer is a standard approach for A3B5 semiconductor integration with sapphire by bonding. However, such an interlayer leads to the high interface state density of ≈6 × 10^11^ cm^−2^eV^−1^ in the case of substitution of the A3B5 layer by silicon [10]. Previously, we proposed and developed methods for inserting thin SiO_2_ and HfO_2_ layers in SOS structures using both silicon thermal oxidation and atomic layer deposition, followed by hydrogen transfer of silicon and annealing of SOS structures, to reduce the large positive charge in the transfer pseudo-MOSFET characteristics [5,6,7]. The inserted additional dielectric layers of silicon and metal dioxides (SiO_2_ and HfO_2_) at the silicon and sapphire boundary prevented internal oxidation and, accordingly, the formation of vacancies in sapphire during high-temperature furnace annealing (FA) of the SOS structure. However, the large hysteresis in the transfer pseudo-MOSFET characteristics was observed for both inserted layers after such FA.

The aim of this study was to establish the hysteresis mechanism using the temperature dependences of the transfer pseudo-MOSFET characteristics. The transport and capture of electrical charges, the recharging of interface states, ion drifts, and ferroelectric polarization in the SOS with inserted dielectrics—silicon and hafnium dioxides—were studied in detail.

The dominant mechanisms of the hysteresis in such structures are analyzed in this investigation with various inserted dielectric layers in the temperature range of 25–300 °C.

## 2. Materials and Methods

The silicon layer transfer from Si (100) wafers with a resistivity of 10–20 Ohm cm (3–5 × 10^14^ cm^−3^) by hydrogen implanted into sapphire was carried out according to the technology [4]. A hafnium dioxide layer with a thickness of 20 nm was grown on some of the wafers before bonding by PEALD using the FlexAl tool (Oxford Instruments Plasma Technology, Yatton, UK). The silicon wafers with a 2 nm native oxide were previously nitrided from a 400 W N_2_ plasma remote ICP source at 500 °C for 5 min in the same tool without breaking the vacuum between the processes of nitridization and HfO_2_ deposition. The organometallic Hf precursor TEMAH (Dalchem, Nizhny Novgorod, Russia) was used to deposit hafnium oxide; the precursor was heated to 70 °C and its vapors were delivered from the bubbler to the chamber by a 250 sccm Ar flow during the 1 s step of each cycle. The remote O_2_ plasma source was used as an oxidizing precursor at a pressure of 15 mTorr and a power of 250 W for 3 s in a cycle. The sample temperature during the PEALD of HfO_2_ was maintained at 250 °C.

Part of the sapphire substrates with the C-orientation were implanted with N^+^ ions with an energy of 50 keV and fluence of 10^16^ cm^−2^ at room temperature. The Si layer transfer was carried out using the method described in Appendix A and earlier in [5,6,7]. Immediately before bonding, the surfaces of a pair of sapphire and silicon wafers were treated in O^+^ or N^+^ plasma. Finally, after the transfer at 450 °C for 1 h, silicon layers with thicknesses of ≈0.5 µm and ≈0.3 µm for thin and thick inserted dielectric layers, respectively, formed the SOS structures. All the obtained SOS wafers were subjected to an FA sequential heat treatment in a high-purity argon atmosphere at temperatures of 800, 1000, and 1100 °C. 

The structural properties and composition of the layers were determined using X-TEM and X-HRTEM transmission electron microscopy, as well as the electron dispersion spectra (EDS) on FEI Titan 80–300 (FEI Company, Hillsboro, OR, USA) and JEM2000FX (JEOL, Ltd Tokyo, Japan) microscopes, respectively (Figure 1). The electric properties of the SOS structures were measured using a home-built high-voltage automatic unit (V_g_ = ±6 kV) from the drain–gate characteristics of pseudo-MOSFETs with tungsten needles at a distance of 100 µm and a tip radius of 20 µm and the clamping force of 60 g as source–drain Schottky barrier contacts in the temperature range of 25–300 °C. Electron and hole mobilities were calculated on the basis of the drain–gate characteristics using the Y-function method (Y = I_DS_/$\sqrt{g_{m}}$) [11,12]:μ_e,h_ = (β_e,h_)^2^/(fC_OX_V_ds_),
where I_ds_ is the drain current, g_m_ is the channel conductivity, β_e,h_ are the Y-function branch slopes for electrons and holes, respectively, f = 0.75 is the geometric factor for two contact measurements, C_OX_ is the gate dielectric capacity, and V_ds_ is the drain voltage, where V_ds_ = 1.5–20 V. 

## 3. Results 

The structural properties and composition of the layers in the SOS structures with the (0001) C-orientation of the 100 mm sapphire substrate were determined using X-TEM, X-HRTEM, and EDS measurements (Figure 1).

Silicon layers, after annealing at temperatures of 1000 °C and higher, are almost free of defects and do not differ in their crystal structure from original bulk silicon. Initially, the amorphous PEALD HfO_2_ layers recrystallize into large-block textured layers, in contrast to the remaining amorphous structure of 50–300 nm SiO_2_ layers [6,7]. Figure 1a shows the distributions of the main elements over the cross-section of such layers, including their distribution maps (Figure 1b). From the X-HRTEM and EDS data, it can be seen that, after high-temperature annealing, a silicon oxide interlayer was formed between the silicon layer and the PEALD inserted HfO_2_ layer. Moreover, aluminum atom diffusion in the hafnium dioxide layer was also observed. Due to the overlap of the Si K and Hf Ma lines, as well as the partial overlap of Al Ka and Hf M and the small (2–3 nm) thickness of the interlayer between silicon and hafnium dioxide, it was not possible to determine the exact composition, although the most likely composition was Hf_x_Si_y_O_z_, which has been experimentally observed many times [13,14].

Measurements of quasi-static transfer (drain–gate I_ds_-V_g_) characteristics of pseudo-MOSFETs were performed using repeated stepwise high gate voltages (−6 kV < V_g_ < 6 kV) changing from the rear contact on the sapphire substrate at a rate of 20–500 V/s relative to the gate voltage V_g,off_, providing a depletion mode for the carriers in the silicon layer (Figure 2, Figure 3 and Figure 4). The gate voltage rate was chosen automatically to satisfy a complete charge relaxation at each voltage step (Appendix A) [15]. The maximum negative or positive voltages corresponded to the hole or electron conductivity in the silicon layer, respectively. Fixed and mobile charge densities in the SOS structures with different inserted dielectric layers were extracted from the drain–gate I_ds_-V_g_ characteristics of pseudo-MOSFET transistors. SOS structures with a SiO_2_ thickness of 310 nm without N^+^ ion implantation (w/o NII) in the sapphire substrates were compared with N^+^-implanted (w NII) SOS structures with a SiO_2_ layer thickness of 50 nm only, with hole and electron branches in the I_ds_-V_g_ range of our high-voltage unit (Figure 2, Figure 3, Figure 4 and Appendix A). The analysis of IV and CV curves allowed for the determination of embedded charge densities in the SOS with inserted dielectrics, which was carried out as follows. The linear extrapolation of the voltage-positive section of the Y-function for a sapphire substrate with a thickness of 70 µm and a HfO_2_ inserted layer provides a threshold voltage V_T,e_ = −1250 ± 100 V and an electron mobility µ_e_ = 230 ± 30 cm^2^/(V⋅s) at V_ds_ = 1.5 V (shown in Figure 3 in [6], but only in one V_g_ direction from −2.5 to +3.0 kV) for the correct mobility measurements, according to [11,12]. The measurements of the I_ds_-V_g_ hysteresis performed in this work at a higher V_ds_ = 10 V (in order to increase the I_ds_ value) in the pseudo-MOSFET structure are presented in Figure 3a. The electron mobility dropped to µ_e_ = 100 cm^2^/(V⋅s) and the drain current saturated. The threshold voltage of holes, V_T,h_ = −1540 ± 100 V, corresponded to the flat band voltage V_FB_. The V_T,e_-V_FB_ difference should not change with a change in the V_ds_ drain voltage and gate dielectric thickness *d* if there are no short-channel effects due to a thick gate dielectric with ε_||_ = 11.5, which gives EOT = (ε_Si_/ε_c-sapp_)⋅d = 0.34⋅d. For sapphires with a thickness of 70 µm, EOT = 2.4 µm, which is much less than the distance between the source and the drain (i.e., 500 µm). Indeed, the experimentally determined threshold voltage value using the slope of the Y-function or the peak of the second derivative of the drain–gate characteristic I_ds_-V_g_ for samples with a thickness of less than 150 µm indicates the threshold voltage independence from the voltage V_ds_ at the drain. Using the slope of the dependence log I_ds_(V_g_) = log I_0_ + V_g_/S at V_g_ < V_T,e_, where the maximum subthreshold slope for electrons S = 255 V/dec, it is possible to estimate the state density D_it_ from the data in Figure 3, Figure 4 and Appendix A using the following formula [11,12]:$$S=2.3\frac{\mathrm{kT}}{q}\left[1+\frac{C_{\mathrm{it}1}}{C_{\mathrm{ox}}}+\frac{C_{\mathrm{Si}}C_{\mathrm{it}2}}{C_{\mathrm{ox}}\left(C_{\mathrm{Si}}+C_{\mathrm{it}2}\right)}\right]$$
where C_it1,2_ = q⋅D_it1,2_ and q is the electron charge. There are capacitances of states on the lower and upper heterogeneous borders of the Si layer. The typical density of broken bonds at the silicon heterogeneous border with the native oxide is D_it2_~2.0 × 10^13^ cm^−2^eV^−1^, but most of them are passivated with hydrogen. Indeed, since C_ox_ is small, C_ox_= C_sa_, and qD_it2_ >> C_Si_ ≈ ε_0_ε_Si_/t_Si_ ≈ 20 nF/cm^2^ in the depletion mode, D_it1_ = D_it(e)_ is equal to
(1)$$D_{\mathrm{it}\left(e,h\right)}=\frac{C_{\mathrm{ox}}}{q}\left[\frac{S_{e,h}}{2.3\frac{\mathrm{kT}}{q}}-\left(1+\frac{C_{\mathrm{Si}}}{C_{\mathrm{ox}}}\right)\right]$$

According to (1), we have D_it(e)_ =7.0 × 10^11^ cm^−2^eV^−1^. The hole mobility µ_h_ in the inversion channel of the pseudo-MOSFET turned out to be significantly less than µ_e_. It was only 35 ± 10 cm^2^/(V s) at V_ds_ = 1.5 V and dropped to µ_h_ = 15 cm^2^/(V⋅s) due to the saturation at V_ds_ = 3 V. A large negative threshold voltage value for electron and hole conductivities, V_T,e_ and V_T,h_ = V_FB_ > 4 kV, for the SOS structures with a thickness of t_sa_ ≥70 µm without inserted dielectric layers corresponded to the capture of a positive charge at the border. The observed positive charge may have been a consequence of the vacancy formation in the SiO_x_ interlayer due to the oxygen atom diffusion into the high-k dielectric [16,17]. The introduction of an intermediate HfO_2_ layer in the SOS partially compensated for this charge and made it possible to roughly estimate the density of states and the effective charge at the heterogeneous interface with silicon using the difference in the threshold voltages of the channels in the enrichment and inversion mode [18]:(2)$$V_{\mathrm{Tn}}-V_{\mathrm{Tp}}≅2\Phi_{F}+\frac{{\mathrm{qt}}_{\mathrm{sa}}}{\varepsilon_{0}\varepsilon_{\mathrm{sa}}}\left(N_{0}t_{\mathrm{Si}}+{2\Phi }_{F}D_{\mathrm{it}}\right)$$
where N_0_ is the donor concentration, Φ_F_ = E_F_ − E_i_ = 0.144 eV is the Fermi level position in the silicon layer bulk with the donor concentration N_D_ = 4 × 10^14^ cm^−3^, C_sa_ = ε_0_⋅ε_||_/t_sa_ is the sapphire capacity, C_sa_ ≈ 113 pF/cm^2^ for 90 µm sapphire (dielectric permittivity ε_sa_ for the field along the axis C ε_||_ = 11.5), D_it_ is the density of states at the interlayer, and t_Si_ is the silicon layer thickness. Then, according to (2), D_it(h)_ = 2.4 × 10^12^ cm^−2^eV^−1^.

The partial depletion mode of the channel made it possible to roughly estimate the positive charge value in the dielectric, reduced to an inserted layer interface with a Si layer as Q_OX_ = −V_FB_C_OX_/q= 1.2 × 10^12^ cm^−2^. Similar calculations carried out for the drain–gate characteristics of SOS structures with a built-in 50 nm silicon dioxide layer at the heterogeneous interface with sapphire, additionally modified by the nitrogen ion implantation, as well as with a 310 nm thick SiO_2_ layer, provided a two times smaller value of D_it_ (Table 1) [6]. A decrease in the positive charge in the presence of a HfO_2_ layer or thermal oxides at the heterogeneous SOS structure interface made it possible to control this charge value, but the D_it_ value turned out to be higher than for the 310 nm thermal SiO_2_ layer (Table 1).

The absence of a depletion mode in the gate voltage operating range in SOS pseudo-MOSFET structures without inserted layers of silicon dioxide or hafnium dioxide is presumably associated with a large positive charge at the silicon–sapphire interface, as well as with an insufficient electric field near the interface due to the thick sapphire substrate. To confirm this assumption, the SOS structures were thinned from the side of the sapphire substrate by grinding and polishing to a thickness value of less than 100 µm, which increased the field strength more than five times without introducing additional defects [5,6]. The measurement of transport properties of carriers using pseudo-MOSFET transistors with a back gate on the substrate side showed normal characteristics of pseudo-MOSFET transistors not only for structures with SiO_2_ layers but also for structures with a 20 nm hafnia layer at the interface with a thin sapphire substrate (Figure 2, Figure 3 and Figure 4). Thinning the substrate allowed pseudo-MOSFET transistors with the SiO_2_ layer to operate in depletion and inversion modes at a bias V_g_ on the substrate of up to ±4 kV.

At the same time, a 20 nm thick hafnia layer provided the appearance of the space charge region of the pseudo-MOSFET, as well as the depletion and inversion modes during the pretreatment of sapphire in nitrogen plasma or nitrogen ion implantation (NII) to compensate for the positive charge, even for 500 µm of the sapphire substrate. In addition, the built-in positive charge at the silicon interface decreased so much that it allowed for the measurements of both electron and hole drain–gate characteristics in the pseudo-MOSFET conducting channel, even with the relatively large sapphire substrate thickness of 150 nm without NII (Figure 3b). Nevertheless, the residual positive charge did not allow for the sapphire thickness of 150 µm to completely repolarize hafnium dioxide with an external field of ≈1 × 10^5^ V/cm in the hole conduction mode, unlike the field of ≈4 × 10^5^ V/cm for the 70 µm substrate thickness (Figure 3a). The measurement results for the drain–gate characteristics of I_ds_-V_g_ pseudo-MOSFETs on sapphire with a built-in SiO_2_ and HfO_2_ dielectric, depending on their temperature, are shown in Figure 4. One interesting feature is the double I_on_/I_off_ ratio increase with temperature increase for the 310 nm thick inserted SiO_2_ layer, while this ratio decreased for the inserted hafnia layer. 

Another relevant factor is the triple increase in the hysteresis window observed for the inserted SiO_2_ layer with rising temperature (Figure 4a), while it was below one-third of the increase for the inserted hafnia layer (Figure 4b). The latter had a thermally stable hysteresis, ΔV_g_ ≈ 600 V, at the SOS pseudo-MOSFET drain–gate characteristics, and this suggests the formation of a ferroelectric phase in the inserted hafnia layer after annealing at 1100 °C, whereas when the charge is captured on traps, the hysteresis bypass loop direction should be the opposite. The HfO_2_ ferroelectric polarization shifted the threshold voltage ΔV_FT_ ≈ 600 V at ±4 kV, which corresponded to a change in the potential ΔV_HfO2_ = ±100 mV and the maximum field of 5 × 10^4^ V/cm in a 20 nm thick HfO_2_ layer. This shift corresponded to a polarization charge of P = ±(80–100) nC/cm^2^ instead of the theoretical value of P = 56 µC/cm^2^ in a field greater than 2 × 10^6^ V/cm, when the orthorhombic phase is most stable [19]. In the measured structure, only part of the hafnia film had ferroelectric properties. The reason is that a further increase in the substrate gate potential was prevented by a surface breakdown. Nevertheless, the polarization field and charge in the hafnium oxide layer can be increased by reducing the sapphire substrate thickness.

Finally, for the various inserted layers, the measurements of transfer (drain–gate I_ds_-V_g_) characteristics and calculations of the mobility of electrons and holes were carried out, wherein the spread of values were 35–50 and 105–250 cm^2^/(Vs), respectively. The built-in charge and interface state density values were (2.1–13) × 10^11^ cm^−2^ and (3.8–24) × 10^11^ cm^−2^eV^−1^, respectively. The polarization charge P was observed to be as low as P = ±(80–100) nC/cm^2^ at the electric field of 5 × 10^4^ V/cm in a 20 nm thick inserted HfO_2_ layer.

## 4. Discussion

The PEALD HfO_2_ layer on silicon usually leads to an increase in the positive charge in the MOSFET dielectric [16], in contrast to the observed decrease in its value in our experiments with high-temperature annealing of SOS. This can be due to different built-in charge formation mechanisms. For example, the negative threshold voltage shift of ΔV_FB_ below 0 V due to the predominance of a positive charge at the silicon/sapphire interface for the SOS structures without preliminary inserted dielectrics may be due to the diffusion of O^2−^ anions from sapphire into silicon dioxide during high-temperature treatments of T > 800 °C [17]. However, the estimation of the diffusion length L based on the volume diffusion coefficient D(T) = 3.27 × 10^−4^ exp(−7.21eV/kT) m^2^/s for the maximum thermal treatment budget (1100 °C during 2 h) provides a too small value of L = 0.01 nm [13]. Another reason for the accelerated diffusion of point defects in this layer may be a chemically reactive fusion border enriched with both vacancies, hydrogen and oxygen atoms. Tangential compression stresses and sapphire tensile stresses normal to the surface can make an additional contribution to the acceleration of diffusion during annealing due to the lower silicon value of CTE. The oxygen atoms exit in planes parallel to the surface, which reduces the misalignment of the lattices during annealing. The probable cause of the large positive charge formation at the silicon/sapphire interface (without inserted dielectrics) is aluminosilicates formed at the SiO_2_/Al_2_O_3_ interface [14]. The inserted HfO_2_ layer reduces the possibility of their formation, which should lead to a decrease in this charge.

It is known that the difference in the chemical bonding polarities of two dielectrics leads to the formation of dipoles at their interface, creating a potential jump. Part of the charges can recombine during dielectric bonding by tunneling through a potential barrier, which leads to a partial loss of the dipole charges [20]. On the other hand, when connecting two flat surfaces of dielectrics (a typical situation), for example, anions are displaced from aluminum oxide, which has a higher surface density of oxygen atoms, into hafnium dioxide, with a 1.37 times lower density [19], leading to the formation of oriented dipoles with a negative charge towards the interface with silicon and the positive shift in the threshold voltage ΔV_FB_ observed in the experiment [21,22]. Experimental results show a negative shift of ΔV_FB_ ≈ −0.4 V for lanthanum oxides with a thickness of 1 nm or more, as well as multiple smaller positive shifts for hafnia and alumina [23].

In our samples, the positive charge at the insulator interface with silicon can be estimated as Q _eff_ = P_p_ − P_n_ = 1.4 × 10^11^ cm^−2^. The repolarization of the inserted HfO_2_ dielectric at room temperature shifted the p- and n-thresholds ΔV_T_ ≈ 600 and −790 V, which corresponded to the electric field 4 × 10^4^ V/cm and polarization charge density (6–7) × 10^11^ cm^−2^ at surface potential change Δϕ_S_ = 80 mV. The ion charge in the inserted SiO_2_ layer demonstrated a quasi-ferroelectric hysteresis, since this hysteresis increases with temperature, and that contradicts the behavior of hysteresis according to the Curie–Weiss law (Figure 4a). The hysteresis memory window (MW) growth at I_ds_ = 20 µA for holes from MW_h_ = 410 V to 1180 V and for electrons from MW_e_ = 550 V to 1280 V with an increase in temperature from 25 to 250 °C was due to the greater diffusion mobility of H^+^ and OH^-^ ions in SiO_2_ [23]. For the inserted HfO_2_ dielectric, the memory window temperature increase was much less pronounced. However, the minimum current I_ds,min_ in the depletion mode of the pseudo-MOSFET transistor grew faster than for the SOS pseudo-MOSFET structure with silicon dioxide (Figure 4b). Therefore, the hysteresis windows were determined for the current difference ΔI_ds_ = I_ds_ − I_ds,min_ = 50 µA with a HfO_2_ layer in the same temperature range. They varied for holes from MW_h_ = 600 V to 680 V and for electrons from MW_e_ = 790 V to 1000 V with a temperature increase from 25 to 250 °C. According to the published data [24], the ion concentration N_ion_ of H^+^ and OH^-^ in the hafnia-based dielectric can reach (1–200) × 10^18^ cm^−3^. Accordingly, the ion current contribution to the charge hysteresis can be estimated as the product of their concentration by the distance l that hydroxyl ions (equivalent to oxygen vacancy) will overcome during the time of pulse signal t with a drift velocity v equal to [25]:(3)$$v=\mathrm{af}\;\exp\left(\frac{-E_{a}}{\mathrm{kT}}\right)\sin h\left(\frac{\mathrm{qaE}}{\mathrm{kT}}\right)\;\mathrm{and}\;l=\mathrm{vt}$$

Here, q is the charge, which, for a proton and hydroxyl, is equal to ±1 elementary charge; a = 0.25 nm is the jump distance; f = 10^12^ Hz is the oxygen atom oscillation frequency; E_a_ = 0.45 eV is the hydroxyl (oxygen vacancies) activation energy of movement; k is the Boltzmann constant; T is the temperature; E = 1 × 10^6^ V/cm is the electric field strength; and t = 200 s is the time until the pulse is applied at the next point of quasi-static drain–gate I_ds_-V_g_ measurements. For these parameters, the drift length l is equal to 0.48 nm at RT and 28 µm at 300 °C. These estimates show that, in a SOS pseudo-MOSFET with a 20 nm HfO_2_ inserted dielectric, the hysteresis at room temperature is associated with ferroelectric repolarization, and a temperature increase of up to 300 °C can collect all ions at their boundaries. The full ion charge is Q_ion_ = N_ion_ d_HfO2_ = 2 × 10^20^ 2 × 10^−6^ = 4 × 10^14^ cm^−2^. Since the polarization measured by the hysteresis window was three orders of magnitude lower, the ion density in the inserted HfO_2_ dielectric was also reduced by more than these three orders and did not exceed N_ion_ ≈ 1 × 10^17^ cm^−3^.

## 5. Conclusions

To reduce the built-in positive charge value at the silicon-on-sapphire (SOS) phase border obtained by bonding and a hydrogen transfer, thermal silicon oxide (SiO_2_) layers with a thickness of 50–310 nm and HfO_2_ layers with a thickness of 20 nm were inserted between silicon and sapphire by plasma-enhanced atomic layer deposition (PEALD). After high-temperature annealing at 1100 °C, these layers led to a hysteresis in the drain current–gate voltage curves and a field-induced switching of threshold voltage in the SOS pseudo-MOSFET. The transfer transistor characteristics measured in the temperature ranging from 25 to 300 °C demonstrated a triple increase in the hysteresis window with increasing temperature for inserted SiO_2_ with a thickness of 310 nm. It was associated with the ion drift and the formation of electric dipoles at the silicon dioxide boundaries. A much slower increase in the window with temperature for the inserted HfO_2_ layer was explained by the dominant ferroelectric polarization switching in the inserted HfO_2_ layer. The experiments allowed for a separation of the effects of mobile ions and ferroelectric polarization on the observed transfer characteristics of hysteresis in Si/HfO_2_/sapphire and Si/SiO_2_/sapphire structures.

The SOS pseudo-MOSFET with the inserted HfO_2_ layer on <150 µm thick sapphire demonstrated normal drain–gate characteristics with charge carrier mobility, as in bulk silicon, as well as a smaller (compared to the SOS structure without an inserted dielectric layer) positive charge value of up to 1 × 10^12^ cm^−2^ and stable ferroelectric-type hysteresis with ΔV_g_ > 600 V. Such characteristics are promising for developing elements of built-in memory and expanding the functionality of integrated SOS circuits for radiophotonics, microwave, and optoelectronics.

The physical reason for the low-field switching in the SOS pseudo-MOSFET with the inserted hafnium dioxide layer is the compressive biaxial stress due to the large CTE difference between silicon and sapphire after heat treatment. The density of ions and charge traps in the HfO_2_ and SiO_2_ layers was found to be small (<5 × 10^11^ cm^−2^) and did not mask the ferroelectric switching in HfO_2_, even in the case of nitrogen-implanted sapphire. Our continued experiments with various thicknesses of inserted hafnia layers in silicon on sapphire yield a potential jump value due to the dipoles at the HfO_2_/SiO_2_ interface.

## Figures and Tables

**Figure 1 nanomaterials-12-03394-f001:**
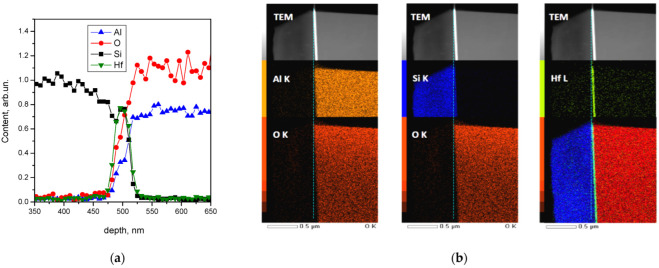
EDS profiles of elements (**a**) and Х-TEM micro-images with X-ray fluorescence maps (**b**) for the SOS cross-section with the 500 nm Si layer and the 20 nm PEALD inserted HfO_2_ layer on the sapphire substrate after annealing at 1100 °C.

**Figure 2 nanomaterials-12-03394-f002:**
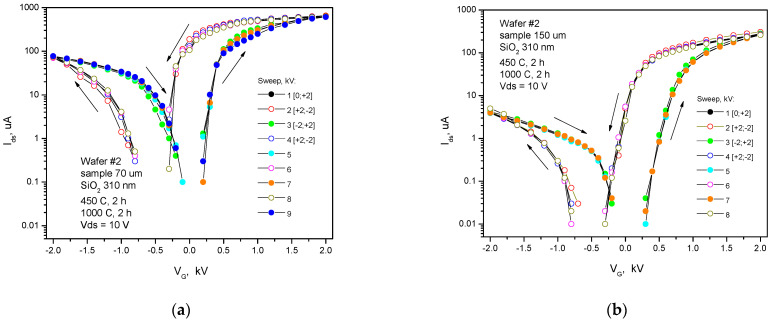
Drain–gate I_ds_-V_g_ transfer characteristics of SOS pseudo-MOSFET structures with a 310 nm thick inserted SiO_2_ layer without N^+^ ion implantation. The sapphire substrates were thinned by grinding to the thicknesses of 70 µm (**а**) and 150 µm (**b**).

**Figure 3 nanomaterials-12-03394-f003:**
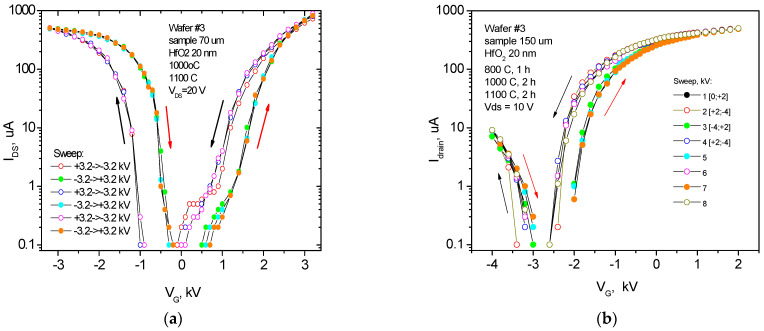
Drain–gate I_ds_-V_g_ transfer characteristics of SOS pseudo-MOSFET structures with a 20 nm thick inserted HfO_2_ layer without N^+^ ion implantation. The sapphire substrates were thinned by grinding to the thicknesses of 70 µm (**а**) and 150 µm (**b**).

**Figure 4 nanomaterials-12-03394-f004:**
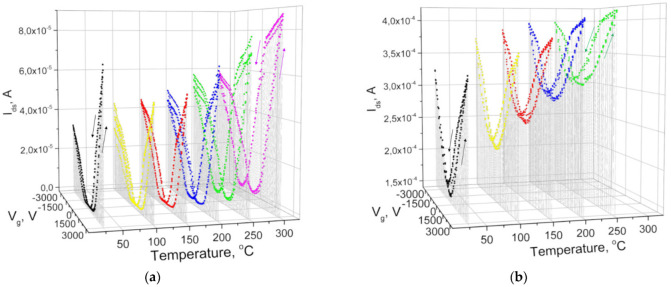
Temperature dependences of pseudo-MOSFET drain–gate I_ds_-V_g_ transfer characteristics for SOS structures with a 310 nm thick inserted SiO_2_ layer (**a**) and 20 nm thick inserted HfO_2_ layer (**b**) without NII after the annealing.

**Table 1 nanomaterials-12-03394-t001:** The values of mobility μ_e_ and μ_h_, built-in charge Q_ox_, and density of states for electrons D_it(e)_ and holes D_it(h)_ at the interface with the transferred Si layer for three types of SOS structures measured at V_ds_ = 1.5 V for the correct mobility measurements according to [11,12,18].

No. and IL Description	μ_e_/μ_h_, cm^2^/(Vs)	Q_ox_, cm^−2^	D_it(e)_/D_it(h)_, cm^−2^eV^−1^
#1 Thin SiO_2_ 50 nm N^+^, 50 keV	105/37	2.1·10^11^	1.3·10^12^/3.8·10^11^
#2 Thick SiO_2_ 310 nm	250/50	4.7·10^11^	6.3·10^11^/4.1·10^11^
#3 Thin HfO_2_ 20 nm	230/35	1.2·10^12^	7.0·10^11^/2.4·10^12^

## Data Availability

Not applicable.

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
