# Peer review of "Ion Drift and Polarization in Thin SiO2 and HfO2 Layers Inserted in Silicon on Sapphire"

_nanomaterials, 2022, doi:10.3390/nano12193394_

Round 1

Reviewer 1 Report

The present paper describes the ion drift and the formation of electric dipoles at the hafnia and silica interfaces in the silicon on sapphire structures. The results are interesting but the paper needs major revision. My suggestions are to extend the introduction describing better the state of the art of the research. The figures quality should be improved together with the overall english. The results should be better described.

Author Response

Answers to the REFEREE REPORT 1

The present paper describes the ion drift and the formation of electric dipoles at the hafnia and silica interfaces in the silicon on sapphire structures. The results are interesting but the paper needs major revision.

My suggestions are to extend the introduction describing better the state of the art of the research.

We corrected the introduction part according to this comment and outlined the new issue of the different hysteresis temperature dependences due to its different nature for the inserted silica and hafnia layers. All corrections are highlighted in the manuscript.

The figures quality should be improved together with the overall English. .

We corrected Figures 2-4 and their descriptions according to this comment.

The results should be better described.

We corrected the Discussion part and explained the results shown in Figures 2-4 in the Results part in the text according to this comment.

Reviewer 2 Report

This manuscript by Popov and co-authors demonstrated that inserting SiO2 and HfO2 layers in the silicon-on-sapphire heterostructure could reduce the built-in positive charge of the SOS interface. The measurement of Ids-Vg characteristics and calculation of electron/hole mobility were carried out for SOS with different intermediate layers. The authors also investigated the mechanism of hysteresis in such structures. Though the story is complete and the conclusion cam be supported by the results, there are many issues with this MS. In my opinion, this manuscript only can be accepted by Nanomaterials after revision.

1.     The English writing of the manuscript is very poor, full of grammar mistakes. The MS will need extensive language revision and improvement.

2.     Small format mistakes have been seen everywhere in the MS, please do proofreading carefully and correct all those small mistakes. For example, we need space between number and unit when we write temperature. The unit of density of state should be cm-2 eV-1, but the shown “cm-2” in the text and so on

3.     The idea presented in this MS is not new, which has been reported in several publications by the same group including https://doi.org/10.1016/j.sse.2019.03.036. The novelty for this MS is not outstanding. The authors need to distinguish this work from their previous work, making the results in this repost standing out.

4.     The explanation on Figure 2 is not clear, the authors doesn’t discuss the result of Figure 2 sufficiently. In addition, after mentioning Figure 4 in line 203, the authors jump to the discussion part without explain the results shown in Figure 4. Then bring back Figure 4 results at the end of Discussion section. This is not logically fluent to me.

Author Response

Answers to the REFEREE REPORT 2

This manuscript by Popov and co-authors demonstrated that inserting SiO2 and HfOlayers in the silicon-on-sapphire heterostructure could reduce the built-in positive charge of the SOS interface. The measurement of Ids-Vg characteristics and calculation of electron/hole mobility were carried out for SOS with different intermediate layers. The authors also investigated the mechanism of hysteresis in such structures. Though the story is complete and the conclusion cam be supported by the results, there are many issues with this MS. In my opinion, this manuscript only can be accepted by Nanomaterials after revision.

  1. The English writing of the manuscript is very poor, full of grammar mistakes. The MS will need extensive language revision and improvement.

We will use the mdpi Editorial office service to avoid language mistakes.

  1. Small format mistakes have been seen everywhere in the MS, please do proofreading carefully and correct all those small mistakes. For example, we need space between number and unit when we write temperature. The unit of density of state should be cm-2eV-1, but the shown “cm-2” in the text and so on

We corrected all found misprints in the text.

  1. The idea presented in this MS is not new, which has been reported in several publications by the same group including https://doi.org/10.1016/j.sse.2019.03.036. The novelty for this MS is not outstanding. The authors need to distinguish this work from their previous work, making the results in this repost standing out.

We corrected the Introduction part according to this comment and outlined the new issue of the different hysteresis temperature dependences due to its different nature for the inserted silica and hafnia layers. All corrections are highlighted in the manuscript.

  1. The explanation on Figure 2 is not clear, the authors doesn’t discuss the result of Figure 2 sufficiently.

We corrected Figures 2-4 and their descriptions according to this comment.

In addition, after mentioning Figure 4 in line 203, the authors jump to the discussion part without explain the results shown in Figure 4. Then bring back Figure 4 results at the end of Discussion section. This is not logically fluent to me.

We corrected the Discussion part and explained the results shown in Figures 2-4 in the Results part in the text according to this comment.

Reviewer 3 Report

   I have general complain about the style and clarity of the text. The authors should improve quality of the text and material presentation. The Abstract should smoothly and clear formulate the research topic and results. There are some disagreements between presented experimental data and their analysis: experimental data and analysis are given for different measurement parameters. The readers should believe to the authors that conclusions are based on the real data. The authors do not to specify at which conditions (value of Vds) they present data in Figures 2-4, and in their analysis they derive device characteristics at Vds range which is not presented in the measurements. It is difficult to follow the author discussion and relay on the obtained results without data which were used for analysis presented on pages 4-5.

   It would be useful if the authors introduce the abbreviations used in the manuscript. For example, it is not clear what means pseudo-MOPT” (or CTE). Is this misprint or something else? There is also opposite examples: abbreviation IL was introduced twice (line 28 and line 126) and only once used in the text. Some introduced abbreviations (for example, short-channel effects (SCE)) are not used in the text. 

   All references should be presented in similar format which required by the journal.

    The usage of the publication which are not available for most of the reader is not acceptable. Ref. [2] in the manuscript cannot be found, but the authors refer to the technology reported in the reference in few places including description of the sample preparation in the current research. Moreover, it is not clear to the readers what is the difference between method suggested in [2] and well developed technologies (example from last century: thttps://patents.justia.com/patent/4775641).

    Some (it is impossible to mention all, I hope the authors will check the manuscript more carefully) examples of the unclear or not well formulated phrases:

There is no information about Vds values for presented plots (Ids vs Vg) in the text and figures caption. The numbers can be found only in some figures. If numbers inside the plots are correct, then what is the reason of the Vds difference for different samples (20V in Fig. 3a and 10Vin case of Fig. 2a,b and Fig. 3b)? Data in Fig. 4a probably measured at Vds=5V and

 p.1, lines 14-15: “the inserted thermal SiO2 layers… are introduced between silicon and sapphire”.

p.1, 15: “These layers provided, after the high-temperature annealing at 1100°C, a field switching…”

 p1, 26-27: ”we have developed a method of implanted hydrogen-induced Si layer transfer…”

 p.1, 27-30: “leads to the growth of a nanometer thin silicon oxide interlayer (IL), formed as a result of oxidation of the transferred silicon device layer, and the formation of oxygen vacancies in sapphire”. Phrase is not perfect and it is not clear which oxigen take part in the mentioned oxidation process and vacancy formations.

 p.1, 37: “which was outside the laboratory measurement unit range.”

 p.2, 63-76. Description of the sample fabrication should be improved and clarified. The authors should consider that readers have no access to Ref. [2]. Probably some parts of the text are missing, for example here (line 74): “after transferring, a silicon layer with the thickness of 0.5 micrometer, All the obtained SOS…”  

 p.2, 92. The author should describe their choice of the gate voltage sweep rates (2 or 500 V/S).

 p.3, 96-98. It is rather strange choice of the structures for comparison: “SOS structures with the SiO2 thickness of 310 nm without N+ ion implantation (w/o NII) in sapphire substrates were used for comparison with 97 N+-implanted (w NII) SOS structures with the SiO2 layer thickness of 50 nm 98”. Could the author comment the reason?

 p.4, Fig.3a. What is a reson for low limit of Ids? Few data sets demonstrate stable constant values around zero gate voltage. Is this measurement limitation or result of higher Vds in case of data in Fig.3a?

 p4, 123: ”In Figure 1a are the distributions…”

 p.4, 132-134: Here readers can find discussion of Y-function behavior, extrapolation results and obtained mobility at Vds=1.5V, but original data are missing. It would be good to present corresponding figure. It is not clear what the authors consider as “electron mobility fμe”: product of mobility and geometric factor or something else.

 P4. 134-136: Rather strange description of the results presented in Fig. 3a: “The measurements of the silicon layer parameters … are presented in Fig. 3a. It drops to 100 cm2/(Vs) with an increase in Vds to 15 V due to the drain current saturation.”

 p.4, 139: “dielectric with e= 11.5 EOT = (eSi/ec-sapp)t = 0.34t).” Strange record, EOT and t are not described.

 p.5, 141: “experimentally determined the threshold voltage value, determined by the slope…”

 p5, 147: Authors should check this equation (last term is not correct) and give better description of the deviation of equation for density od states.

 P5, 150-151: “Indeed, since Cox is small, since Cox= Csa, and…”

 P5, 164: “where q is the electron charge”.  q was used before it was introduced here (see, for example, line 147, 148, 153).

 p. 5, Eq. 2: It is not clear how the authors derived this equation and the basis for used parameter in following analysis.

 p.6, 184-186: “To verify this assumption, the SOS structures were thinned from the side of the sapphire substrate by grinding and polishing to a thickness value of less than 100 micrometers”.

The mechanical treatment of substrate affects the quality of the devices by introducing additional defects which may lead to degrade of the device mobility and increase or decrease of the interface charge. Have these effects been analyzed/investigated?

P. 7, Fig 4 and discussion on p. 8: The information about gate voltage sweeping rate is missing. The observed hysteresis can be described both by capacitance of the sample and parasitic capacitance in measurement setup. Obviously, data for few sweeping rates will be very useful.

Author Response

Answers to the REFEREE REPORT 3

   I have general complain about the style and clarity of the text.

The authors should improve quality of the text and material presentation.

The Abstract should smoothly and clear formulate the research topic and results.

There are some disagreements between presented experimental data and their analysis: experimental data and analysis are given for different measurement parameters. The readers should believe to the authors that conclusions are based on the real data. The authors do not to specify at which conditions (value of Vds) they present data in Figures 2-4, and in their analysis they derive device characteristics at Vds range which is not presented in the measurements. It is difficult to follow the author discussion and relay on the obtained results without data which were used for analysis presented on pages 4-5.

We specified the conditions of Vds values in Figures 2-4 that correspond to the whole Ids measurement range (0.1-1000 µA) of our home-built high voltage unit, while we derived mainly the hysteresis window in the work, beside Tab.1. where Vds value were only 1.5 V for the mobility correct measurements according to [9, 10, 16]. Other Vds values were presented in our previous publications [3-5, 13]. We corrected the sentences according to this comment in the text: “…The Vds values were chosen automatically to correspond to the whole Ids measurement range (0.1-1000 µA) during the gate voltage Vg sweep in the home-built high voltage unit. The gate voltage rate was corrected also automatically to keep a current relaxation value <5% at each voltage step (Fig. S2b, Supplementary materials) [13].”

   It would be useful if the authors introduce the abbreviations used in the manuscript. For example, it is not clear what means ”pseudo-MOPT” (or CTE). Is this misprint or something else?  

We corrected the sentences in the text according to this comment. All corrections are highlighted in the manuscript.

There is also opposite examples: abbreviation IL was introduced twice (line 28 and line 126) and only once used in the text. Some introduced abbreviations (for example, short-channel effects (SCE)) are not used in the text. 

We corrected the sentence in the text according to this comment.

   All references should be presented in similar format which required by the journal.

We corrected all references in the text according to this comment.

    The usage of the publication which are not available for most of the reader is not acceptable. Ref. [2] in the manuscript cannot be found, but the authors refer to the technology reported in the reference in few places including description of the sample preparation in the current research. Moreover, it is not clear to the readers what is the difference between method suggested in [2] and well developed technologies (example from last century: thttps://patents.justia.com/patent/4775641).

We corrected the sentences according to this comment in the text: “To overcome this obstacle, we have developed a method similar to SmartCut© by implanted hydrogen-induced Si layer transfer in vacuum at the elevated temperature bonding [2].”

We changed the reference [2] by I. E. Tyschenko, E. D. Zhanaev & V. P. Popov. Bonding Energy of Silicon and Sapphire Wafers at Elevated Temperatures of Joining. Semicond. 2019, 53, 60–64.

    Some (it is impossible to mention all, I hope the authors will check the manuscript more carefully) examples of the unclear or not well formulated phrases:

There is no information about Vds values for presented plots (Ids vs Vg) in the text and figures caption. The numbers can be found only in some figures. If numbers inside the plots are correct, then what is the reason of the Vds difference for different samples (20V in Fig. 3a and 10Vin case of Fig. 2a,b and Fig. 3b)? Data in Fig. 4a probably measured at Vds=5V and

We corrected all Figure captures according to this comment. All corrections are highlighted in the manuscript.

 p.1, lines 14-15: “the inserted thermal SiO2 layers… are introduced between silicon and sapphire”.

We corrected the misprint according to this comment in the text: “the inserted thermal SiO2 layers …are placed between silicon and sapphire similar to the SmartCut© transfer of both Si and SiO2 layers.”

p.1, 15: “These layers provided, after the high-temperature annealing at 1100°C, a field switching…”

We corrected the misprint according to this comment in the text: “These layers provided, after the high-temperature annealing at 1100°C, a gate electric field induced switching …”

 p1, 26-27: ”we have developed a method of implanted hydrogen-induced Si layer transfer…”

We corrected the misprints according to this comment in the text: ”…we have developed a method of implanted hydrogen-induced Si layer transfer, similar to the SmartCut© transfer,…”

 p.1, 27-30: “leads to the growth of a nanometer thin silicon oxide interlayer (IL), formed as a result of oxidation of the transferred silicon device layer, and the formation of oxygen vacancies in sapphire”. Phrase is not perfect and it is not clear which oxygen take part in the mentioned oxidation process and vacancy formations.

We corrected the misprint according to this comment in the text: “…leads to the growth of a nanometer thin silicon oxide interlayer, formed as a result of silicon oxidation by the decomposed water molecules at the bonding interface and the alumina reduсtion by the silicon atoms.

 p.1, 37: “which was outside the laboratory measurement unit range.”

We corrected the misprints according to this comment in the text: ”… which was outside  the programmable bias voltage range Vg = ±6 kV.”

 p.2, 63-76. Description of the sample fabrication should be improved and clarified. The authors should consider that readers have no access to Ref. [2]. Probably some parts of the text are missing, for example here (line 74): “after transferring, a silicon layer with the thickness of 0.5 micrometer, All the obtained SOS…”  

We corrected the sentences according to this comment in the text: “Immediately before bonding, the surfaces of a pair of sapphire and silicon wafers were treated in O+ or N+ plasma. Finally, after the transfer at 450oC during 1 hour, silicon layers with the thickness of ~0.5 micrometer for the thin and ~0.3 µm for the thick I, respectively, were placed in the SOS structures.

 p.2, 92. The author should describe their choice of the gate voltage sweep rates (2 or 500 V/S).

We corrected the sentences according to this comment in the text: “… at a rate of 20 - 500 V/s relative to the gate voltage Vg,off providing a depletion mode for the carriers in the silicon layer (Fig. 2 - 4). The gate voltage rate was corrected also automatically to keep a current relaxation value <5% at each voltage step (Fig. S2b, Supplementary materials) [13].”

 p.3, 96-98. It is rather strange choice of the structures for comparison: “SOS structures with the SiO2 thickness of 310 nm without N+ ion implantation (w/o NII) in sapphire substrates were used for comparison with 97 N+-implanted (w NII) SOS structures with the SiO2 layer thickness of 50 nm 98”. Could the author comment the reason?

We corrected the sentences according to this comment in the text: “…only both with the hole and electron branches in the Ids-Vg range of our high voltage unit (Fig. 2-4, S2a, Supplementary Materials).

 p.4, Fig.3a. What is a reason for low limit of Ids? Few data sets demonstrate stable constant values around zero gate voltage. Is this measurement limitation or result of higher Vds in case of data in Fig.3a?

Yes. It is simply the measurement limit of our home-built high voltage unit. We corrected Fig. 3a according to this comment in the text.

 p4, 123: ”In Figure 1a are the distributions…”

What is wrong here?

 p.4, 132-134: Here readers can find discussion of Y-function behavior, extrapolation results and obtained mobility at Vds=1.5V, but original data are missing. It would be good to present corresponding figure. It is not clear what the authors consider as “electron mobility fμe”: product of mobility and geometric factor or something else.

We corrected the misprint “fμe” according to this comment in the text. We corrected also the sentences according to this comment in the text as: “The linear extrapolation of the voltage-positive section of the Y-function for a sapphire substrate with the thickness of 70 µm HfO2 IL gives a threshold voltage VT,e = -1250±100 V and an electron mobility µe = 230±30 cm2/(V×s) at Vds = 1.5 V (shown in Fig. 3 in [4], but only in one Vg direction from -2.5 to +3.0 kV). The measurements of the Ids-Vg hysteresis at higher Vds = 10 V, in order to increase the Ids value in the pseudo-MOSFET structure, are presented in Fig. 3a.”

 P4. 134-136: Rather strange description of the results presented in Fig. 3a: “The measurements of the silicon layer parameters … are presented in Fig. 3a. It drops to 100 cm2/(Vs) with an increase in Vds to 15 V due to the drain current saturation.”

We corrected the sentences according to this comment in the text as: “The linear extrapolation of the voltage-positive section of the Y-function for a sapphire substrate with the thickness of 70 µm HfO2 IL gives a threshold voltage VT,e = -1250±100 V and an electron mobility µe = 230±30 cm2/(V×s) at Vds = 1.5 V (shown in Fig. 3 in [4], but only in one Vg direction from -2.5 to +3.0 kV). The measurements of the Ids-Vg hysteresis at higher Vds = 10 V in order to increase the Ids value in the pseudo-MOSFET structure are presented in Fig. 3a. The electron mobility drops to µe = 100 cm2/(V×s) due to the drain current saturation.”

 p.4, 139: “dielectric with e= 11.5 EOT = (eSi/ec-sapp)×= 0.34×t).” Strange record, EOT and t are not described.

We changed “t” for “d” described d in the upper sentence and corrected the sentence according to this comment in the text as: “The VT,e-VFB difference should not change with a change in the Vds drain voltage and thickness if there are no short-channel effects due to a thick gate dielectric with e|| = 11.5 EOT = (eSi/ec-sapp)×d = 0.34×d.”

 p.5, 141: “experimentally determined the threshold voltage value, determined by the slope…”

We corrected the sentence according to this comment in the text as: “For sapphire with the thickness of 70 µm, EOT = 2.4 µm, which is much less than the distance between the source and the drain of 500 µm. Indeed, the experimentally determined threshold voltage value by the slope of the Y-function …”

 p5, 147: Authors should check this equation (last term is not correct) and give better description of the deviation of equation for density of states.

We corrected this equation in the text according to the comment.

 P5, 150-151: “Indeed, since Cox is small, since Cox= Csa, and…”

We corrected the sentence according to this comment in the text as: “Indeed, since Cox is small Cox= Csa, and…”

 P5, 164: “where q is the electron charge”.  q was used before it was introduced here (see, for example, line 147, 148, 153).

We corrected the misprints according to this comment in the text.

  1. 5, Eq. 2: It is not clear how the authors derived this equation and the basis for used parameter in following analysis.

We used this equation from [16] applying the same parameters as in [3, 4] and corrected the misprint according to this comment in the text.

 p.6, 184-186: “To verify this assumption, the SOS structures were thinned from the side of the sapphire substrate by grinding and polishing to a thickness value of less than 100 micrometers”.

The mechanical treatment of substrate affects the quality of the devices by introducing additional defects which may lead to degrade of the device mobility and increase or decrease of the interface charge. Have these effects been analyzed/investigated?

We studied this question in [3, 4]. We did not observe (Fig.6 in [4]) the Vfb voltage change in the thick silica IL case during the sapphire thinning and more complicated behavior was observed for thin silica and hafnia ILs with different changes. We corrected the sentence according to this comment in the text as: “To verify this assumption, the SOS structures were thinned from the side of the sapphire substrate by grinding and polishing to a thickness value of less than 100 micrometers, which increased the field strength more than 5 times without an additional defect introduction [3, 4].”

  1. 7, Fig 4 and discussion on p. 8: The information about gate voltage sweeping rate is missing. The observed hysteresis can be described both by capacitance of the sample and parasitic capacitance in measurement setup. Obviously, data for few sweeping rates will be very useful.

We used quasi-static regimes for the drain current measurement at the constant gate voltage typically during 4 s at each step for Fig.4 (see SM and Fig. 1, 2 in [13]), It is the reason why the parasitic capacitance measurement is not present in this regime.

Round 2

Reviewer 1 Report

The manuscript improved but still needs some language editing.

Author Response

Below our answers to the reviewer 1:

The manuscript improved but still needs some language editing.

  1. The Manuscript was edited by the MDPI English proofreading service.
  2. We added new references [2, 3] and sentences in Introduction according to the remark:
    “There are few reports for producing SOS structures using the hydrogen induced thin Si layer transfer on sapphire by a wafer bonding [3-7]. Usually, the bonding with sapphire is used for nitride semiconductor or niobate dielectric wafers with the similar to sapphire coefficient of thermal expansion (CTE). To overcome the CTE difference the authors [3] used the local fast laser heating to diminish the full stresses in the bonded wafers and only the partial Si layer transfer. Instead, we suggested in [4-7] the silicon and sapphire wafer bonding at the moderate temperature and the followed hydrogen induced thin Si layer transfer on the whole sapphire wafer at the elevated temperature.”
  3. We added new references [2, 3].
  4. We change the Conclusion according to the remark

Reviewer 3 Report

The manuscript has been revised according recommendations. New version is characterized by smother and clear result presentation and discussion, most of unclear/missing statements have been corrected. The manuscript can be considered for publication after minor corrections. New variant of Fig. 4 is more illustrative but has technical problems: font sizes should be similar and easy to read for all axes (too small for Vg and too large and overlapped for Temperature).

Author Response

The manuscript has been revised according recommendations. New version is characterized by smother and clear result presentation and discussion, most of unclear/missing statements have been corrected. The manuscript can be considered for publication after minor corrections. New variant of Fig. 4 is more illustrative but has technical problems: font sizes should be similar and easy to read for all axes (too small for Vg and too large and overlapped for Temperature).

  1. There are few reports for the SOS structure producing using the hydrogen induced thin Si layer transfer on sapphire by a wafer bonding [3-7]. Usually, the bonding with sapphire is used for nitride semiconductor or niobate dielectric wafers with the similar to sapphire coefficient of thermal expansion (CTE). To overcome the CTE difference the authors [3] used the local fast laser heating to diminish the full stresses in the bonded wafers and only the partial Si layer transfer was obtained. Instead, we suggested in [4-7] the silicon and sapphire wafer bonding at the moderate temperature and the followed hydrogen induced thin Si layer transfer on the whole wafer at the further elevated temperature.
  2. We did not find other reports besides [3-7] for producing SOS wafers using the hydrogen induced thin Si layer transfer on sapphire by wafer bonding. The methods were described completely in our previous reports [4-7].
  3. We changed graphs and axes value fonts in Figure 4 according to the comment.
  4. We changed Conclusion according to the remark
